# β-blockers augment L-type Ca$^{2+}$ channel activity by targeting spatially restricted β$_2$AR signaling in neurons

**Ao Shen[1,2], Dana Chen[2], Manpreet Kaur[2], Peter Bartels[2], Bing Xu[2,3], Qian Shi[2], Joseph M Martinez[2], Kwun-nok Mimi Man[2], Madeline Nieves-Cintron[2], Johannes W Hell[2], Manuel F Navedo[2], Xi-Yong Yu[1], Yang K Xiang[2,3]\***

[1]Key Laboratory of Molecular Target and Clinical Pharmacology, State Key Laboratory of Respiratory Disease, School of Pharmaceutical Sciences & the Fifth Affiliated Hospital, Guangzhou Medical University, Guangzhou, China; [2]Department of Pharmacology, University of California Davis, Davis, United States; [3]VA Northern California Health Care System, Mather, United States

**Abstract** G protein-coupled receptors (GPCRs) transduce pleiotropic intracellular signals in mammalian cells. Here, we report neuronal excitability of β-blockers carvedilol and alprenolol at clinically relevant nanomolar concentrations. Carvedilol and alprenolol activate β$_2$AR, which promote G protein signaling and cAMP/PKA activities without action of G protein receptor kinases (GRKs). The cAMP/PKA activities are restricted within the immediate vicinity of activated β$_2$AR, leading to selectively enhance PKA-dependent phosphorylation and stimulation of endogenous L-type calcium channel (LTCC) but not AMPA receptor in rat hippocampal neurons. Moreover, we have engineered a mutant β$_2$AR that lacks the catecholamine binding pocket. This mutant is preferentially activated by carvedilol but not the orthosteric agonist isoproterenol. Carvedilol activates the mutant β$_2$AR in mouse hippocampal neurons augmenting LTCC activity through cAMP/PKA signaling. Together, our study identifies a mechanism by which β-blocker-dependent activation of GPCRs promotes spatially restricted cAMP/PKA signaling to selectively target membrane downstream effectors such as LTCC in neurons.

**\*For correspondence:**
ykxiang@ucdavis.edu

**Competing interests:** The authors declare that no competing interests exist.

## Introduction

GPCRs often signal not only through canonical G proteins but also through noncanonical G protein-independent signaling, frequently via G protein receptor kinases (GRKs) and β-arrestins (*Lefkowitz, 2000*; *Xiang and Kobilka, 2003*). One of the universal features of GPCRs is that they undergo ligand-induced phosphorylation at different sites by either GRKs or second messenger dependent protein kinases such as protein kinase A (PKA). The phosphorylated GPCRs thus may present distinct structural features that favor receptor binding to different signaling partners, engaging distinct downstream signaling cascades (*Reiter and Lefkowitz, 2006*; *Lefkowitz, 2007*; *Nobles et al., 2011*). Some ligands can differentially activate a GPCR via a phenomenon known as functional selectivity or biased signaling (*Wisler et al., 2014*; *Zweemer et al., 2014*). For example, stimulation of β$_2$ adrenergic receptor (β$_2$AR), a prototypical GPCR involved in memory and learning in the central nervous system (CNS) and regulation of metabolism and cardiovascular function, promotes phosphorylation by both GRKs and PKA (*Najafi et al., 2016*; *Mammarella et al., 2016*; *Fu et al., 2017*; *Matera et al., 2018*). We have recently identified spatially segregated subpopulations of β$_2$AR undergoing exclusive phosphorylation by GRKs or PKA in a single cell. These findings indicate specific GPCR subpopulation-based signaling branches can co-exist in a single cell (*Shen et al., 2018*). GRK-mediated phosphorylation promotes pro-survival and cell growth signaling via β-arrestin-

dependent mitogen-activated protein kinase (MAPK/ERK) pathways, prompting the search for biased ligands that preferentially activate β-arrestin pathways (*Luttrell et al., 1999*; *Pierce et al., 2000*; *Kim et al., 2005*; *Ren et al., 2005*; *Zidar et al., 2009*; *Choi et al., 2018*). On the other hand, our recent studies show that the cAMP/PKA-dependent phosphorylation of β₂AR controls ion channel activity at the plasma membrane in primary hippocampal neurons (*Shen et al., 2018*).

β-blockers are thought to reduce cAMP signaling because they either reduce basal activity of βARs or block agonist-induced receptor activation. While β-blockers are successful in clinical therapies of a broad range of diseases, their utility is limited by side effects in both the CNS and peripheral tissues (*Bakris, 2009*; *Gorre and Vandekerckhove, 2010*). Indeed, studies have revealed that some β-blockers display partial agonism and can promote receptor-Gs coupling at high concentrations in vitro (*Yao et al., 2009*; *DeVree et al., 2016*; *Gregorio et al., 2017*). Accordingly, some β-blockers display intrinsic properties mimicking sympathetic activation (sympathomimetic β-blockers) (*Maack et al., 2000*; *Brixius et al., 2001*; *Larochelle et al., 2014*). The mechanism remains poorly understood because classic cAMP assay do not show even minimal cAMP signal induced by these β-blockers (*Maack et al., 2000*; *Brixius et al., 2001*).

In this study, we show that the β-blockers carvedilol and alprenolol can promote Gs protein coupling to β₂AR and cAMP/PKA but not GRK activity at nanomolar concentrations. Thus, these β-blockers are emerging as partial agonists even at low concentrations rather than strict antagonists in mammalian cells. This cAMP/PKA signaling is spatially restricted, selectively promoting phosphorylation of β₂AR and Ca$_V$1.2 by PKA which augments LTCC activity in primary hippocampal neurons. Furthermore, we have engineered a mutant β₂AR that can be selectively activated by carvedilol but not by the orthosteric agonist isoproterenol (ISO) to stimulate PKA but not GRK. Together, these studies identify a unique mechanism by which β-blockers activate β₂AR at low concentrations, which promotes Gs/cAMP/PKA signaling branch and selectively targets downstream LTCC channels in neurons. This observation may also explain sympathomimetic effects of β-blockers in the CNS.

## Results

### Carvedilol and alprenolol selectively promote β₂AR-mediated PKA-phosphorylation of β₂AR

In this study, we applied two sets of well-characterized phospho-specific antibodies, anti-pS261/262 and anti-pS355/356 to examine a series of β-blockers for their effects on the phosphorylation of β₂AR at its PKA and GRKs sites, respectively (*Shen et al., 2018*; *Tran et al., 2004*; *Tran et al., 2007*). We found that various β-blockers including alprenolol (ALP), carvedilol (CAR), propranolol (PRO) and CGP12177 (177) were able to stimulate phosphorylation of β₂AR at PKA sites expressed in HEK293 cells, similar to the βAR agonist isoproterenol (ISO) (*Figure 1A* and *Figure 1—figure supplement 1A*). In contrast, other β-blockers, that is ICI118551 (ICI), timolol (TIM) and metoprolol (MET), were not able to do so (*Figure 1A*). The ligand-induced phosphorylation of β₂AR was blocked by β₂AR-specific antagonist ICI but not β₁AR-specific antagonist CGP20712A (CGP) (*Figure 1B and C*, and *Figure 1—figure supplement 1B and C*). We chose CAR and ALP for further study. We found that CAR and ALP promoted phosphorylation of β₂AR by PKA even at nanomolar concentrations (*Figure 2A and B*, and *Figure 2—figure supplement 1A and B*), which was paralleled by concentration-dependent increases in phosphorylation of ERK (*Figure 2—figure supplement 2*). The roles of β₂AR and PKA in this phenomenon were confirmed by inhibition of β₂AR with ICI and inhibition of PKA with H89, respectively (*Figure 2C and D*, and *Figure 2—figure supplement 1C and D*). In contrast, those β-blockers induced at best minimal increases in phosphorylation of β₂AR at GRK sites and only at high concentrations, consistent with a previous report (*Wisler et al., 2007*) (*Figure 1A* and *Figure 2—figure supplement 3*). As positive control, the βAR agonist ISO promoted robust increases in both PKA and GRK phosphorylation of the receptors at different concentrations ranging from nanomolar to micromolar (*Figures 1* and *2*, and *Figure 2—figure supplements 2* and *3*). In the CNS, β₂AR emerges as a prevalent postsynaptic norepinephrine effector at glutamatergic synapses (*Davare et al., 2001*; *Joiner et al., 2010*; *Wang et al., 2010*; *Qian et al., 2012*). Consistent with the data from HEK293 cells, we found β-blockers CAR and ALP activated β₂AR and promoted phosphorylation of the receptor by PKA in hippocampal neurons (*Figure 2E*). Together,

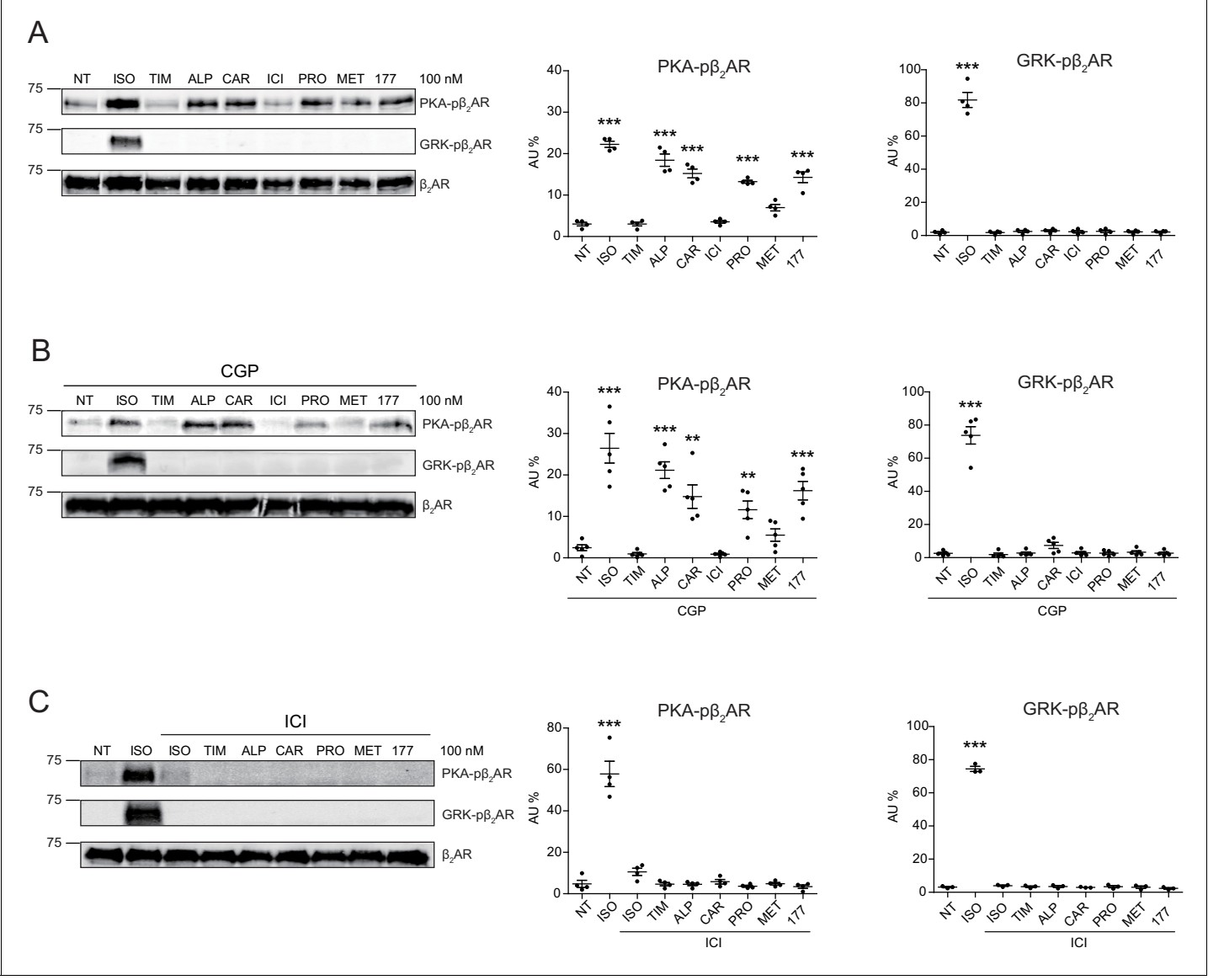

**Figure 1.** Carvedilol and alprenolol selectively promote phosphorylation of $\beta_2AR$ at PKA sites. HEK293 cells stably expressing FLAG-tagged $\beta_2AR$ were either directly stimulated for 5 min with the $\beta AR$ agonist ISO or different $\beta$-blockers at indicated concentrations (**A**) n = 4, or pretreated for 15 min with 1 $\mu$M $\beta_1AR$ antagonist CGP20712A (**B**) n = 5) or 10 $\mu$M $\beta_2AR$ antagonist ICI118551 (**C**) n = 4) before the treatment. The phosphorylation of $\beta_2AR$ on its PKA and GRK sites were determined with phospho-specific antibodies, and signals were normalized to total $\beta_2AR$ detected with anti-FLAG antibody. NT, no treatment; ISO, isoproterenol; TIM, timolol; ALP, alprenolol; CAR, carvedilol; ICI, ICI118551; PRO, propranolol; MET, metoprolol; 177, CGP12177; CGP, CGP20712A. Error bars denote s.e.m., *P* values are computed by one-way ANOVA followed by Tukey's test between NT and other groups.

The online version of this article includes the following source data and figure supplement(s) for figure 1:

**Source data 1.** Excel spreadsheet containing the individual numeric values of phosphorylated $\beta_2AR$ / total $\beta_2AR$ relative density analyzed in *Figure 1*.
**Figure supplement 1.** Uncropped blots for *Figure 1*.

these data suggest that certain $\beta$-blockers selectively promote PKA phosphorylation of $\beta_2AR$ in HEK293 and primary hippocampal neurons.

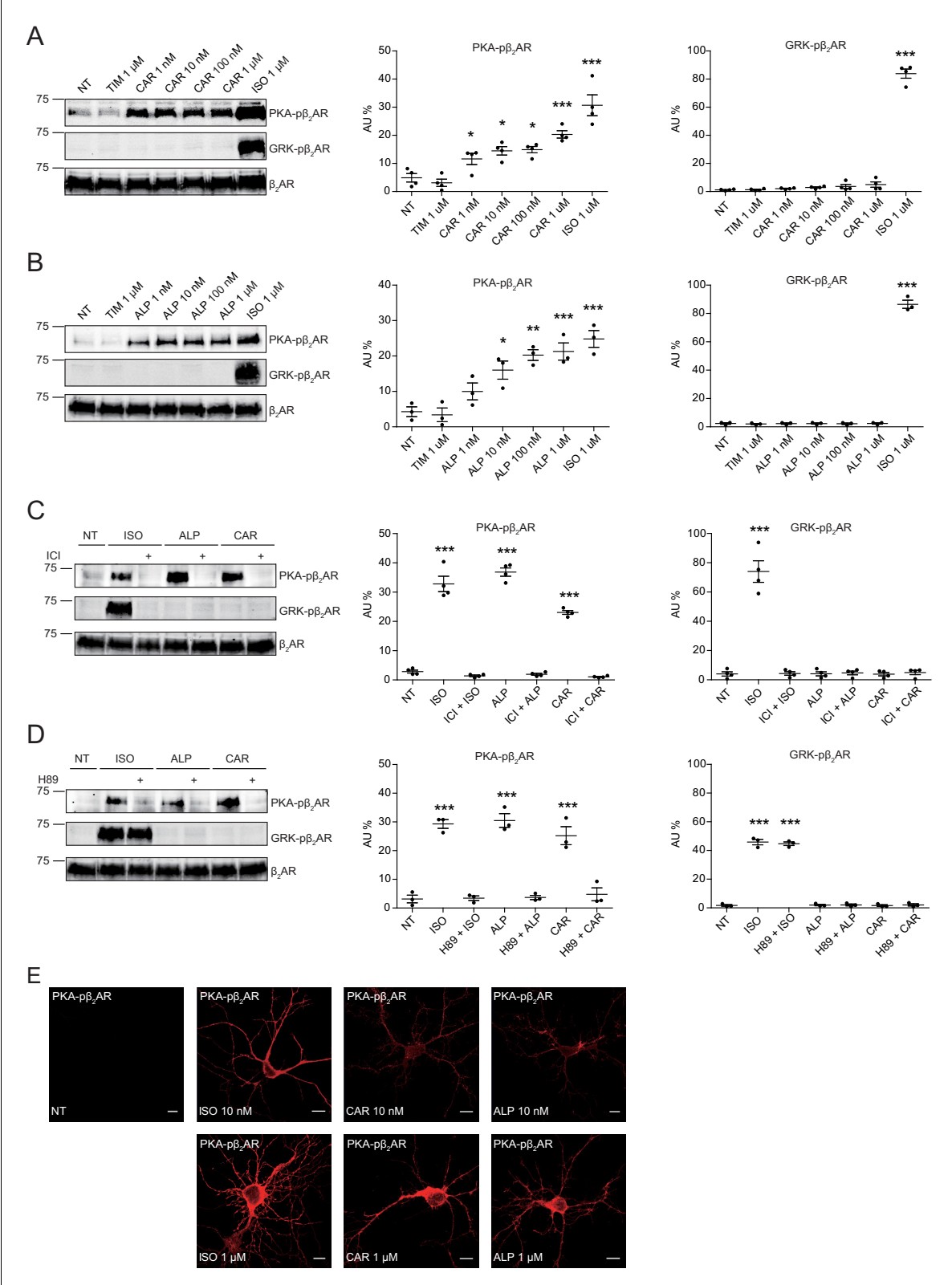

**Figure 2.** Carvedilol and alprenolol induce concentration-dependent PKA phosphorylation of $\beta_2AR$ in HEK293 and hippocampal neurons. HEK293 cells stably expressing FLAG-tagged $\beta_2AR$ were treated with increasing concentrations of CAR (**A**) n = 4) and ALP (**B**) n = 3), or pretreated for 15 min with 10 $\mu$M $\beta_2AR$ antagonist ICI118551 (**C**) n = 4) and PKA inhibitor H89 (**D**) n = 3) before stimulated with 1 $\mu$M indicated drugs for 5 min. The phosphorylation of $\beta_2AR$ on its PKA and GRK sites were determined with phospho-specific antibodies, and signals were normalized to total $\beta_2AR$ detected with anti-

*Figure 2 continued on next page*

Figure 2 continued

FLAG antibody. Experiments were performed in the presence of 1 µM β₁AR-selective antagonist CGP20712A to block endogenous β₁AR signaling. NT, no treatment; ISO, isoproterenol; ALP, alprenolol; CAR, carvedilol; ICI, ICI118551. Error bars denote s.e.m., $P$ values are computed by one-way ANOVA followed by Tukey's test between NT and other groups. (E) Rat hippocampal neurons expressing β₂AR were treated for 5 min with 10 nM or 1 µM indicated drugs on 12 days in vitro (DIV), and immuno-stained for PKA-phosphorylated β₂AR. Confocal images show PKA-phosphorylated β₂AR in agonist- or β-blocker-stimulated neurons have similar distribution. Scale bar, 10 µm. Representative of 6 images for each condition, three experiments.

The online version of this article includes the following source data and figure supplement(s) for figure 2:

**Source data 1.** Excel spreadsheet containing the individual numeric values of phosphorylated β₂AR / total β₂AR relative density analyzed in *Figure 2A-D*.
**Figure supplement 1.** Uncropped blots for *Figure 2*.
**Figure supplement 2.** Phosphorylation of ERK and β₂AR at different drug concentrations.
**Figure supplement 3.** GRK-phosphorylation of β₂AR at different carvedilol treated times.

## Carvedilol and alprenolol promote Gsα recruitment to β₂AR and increase spatially restricted cAMP signal

The western blot data on PKA phosphorylation of β₂AR indicates a stimulation of the receptor-mediated Gs/AC/cAMP pathway by these β-blockers. We measured ligand-induced Gsα recruitment to β₂AR with an in situ proximity ligation assay (PLA), which allows direct visualization and quantification of protein-protein interactions. We showed that ISO, CAR and ALP were able to increase the PLA signals between β₂AR and Gsα, indicating recruitment of Gsα to β₂AR (*Figure 3A*). As control, TIM had no effect on the recruitment of Gsα to β₂ARs. The role of Gs/AC in CAR-induced PKA phosphorylation of β₂AR was further validated by AC-specific inhibition with 2′,5′-dideoxyadenosine (ddA, *Figure 3—figure supplement 1*). These data indicate that CAR and ALP are able to stimulate β₂AR-Gs signal to increase PKA phosphorylation of the receptor.

β-blockers have been thought to generally block β₂AR-induced cAMP signal. We hypothesized that the cAMP signal induced by β-blockers is restricted to local plasma membrane domains containing activated receptor, which is not detectable with traditional cAMP assays likely due to limited sensitivity. We applied the highly sensitive FRET-based biosensor ICUE3 to detect the dynamics of cAMP signal in living cells (*DiPilato and Zhang, 2009*; *De Arcangelis et al., 2009*). The full agonist ISO promoted cAMP signal in HEK293 cells while all β-blockers failed to do so (*Figure 3B*), in agreement with the classic definition of β-blockers. However, when cells were treated with non-selective phosphodiesterase (PDE) inhibitor IBMX, CAR, ALP and CGP12177 were able to induce small but significant cAMP signal in HEK293 cells (*Figure 3C*), indicating a role of PDE in suppressing and restricting the distribution of cAMP in the cells. When β₂AR was exogenously expressed in HEK293 cells, CAR and ALP were able to induce cAMP signal in HEK293 cells even without PDE inhibition (*Figure 3—figure supplement 2*), probably due to insufficient cAMP-hydrolytic activity of endogenous PDEs to counter cAMP production induced from overexpressed β₂AR. We then engineered a targeted cAMP biosensor by fusing the biosensor ICUE3 to the C-terminus of β₂AR (β₂AR-ICUE3), aiming to detect increases of cAMP within the local domain of the receptor. CAR and ALP promoted cAMP signals within the immediate vicinity of activated β₂AR even at nanomolar concentrations (*Figure 3D and E*). The local increases of cAMP were abolished by inhibition of β₂AR with ICI or inhibition of ACs with ddA (*Figure 3E*). We also used two generic plasma membrane (PM) targeted ICUE3 sensors to further characterize how the CAR and ALP generated cAMP signals are localized when compared to the full agonist ISO. Interestingly, neither CAAX-ICUE3 targeting to the non-rafts regions of PM nor LYN-ICUE3 targeting to the rafts regions of PM could sense cAMP induced by CAR and ALP, while ISO induced cAMP were readily detectable on PM (*Figure 3—figure supplement 3*), this further demonstrates that CAR and ALP only promote cAMP within the immediate vicinity of β₂AR. These data confirm that CAR and ALP promote cAMP/PKA activity within the immediate vicinity of activated β₂AR, in contrast to the broad distribution of cAMP/PKA activities induced by ISO in the cells.

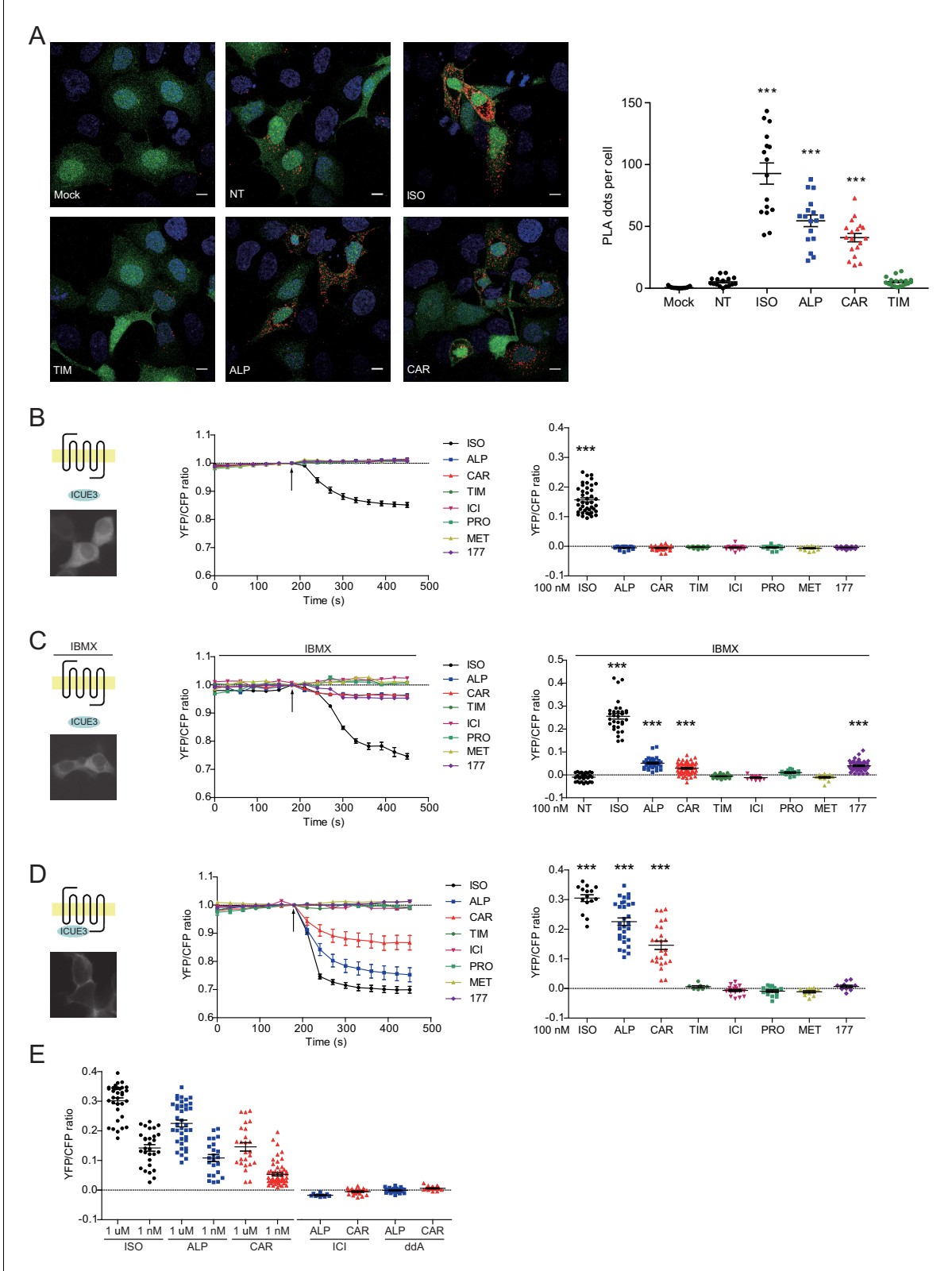

**Figure 3.** Carvedilol and alprenolol promote Gsα recruitment to β2AR and increase spatially restricted cAMP signal. (**A**) HEK293 cells co-expressing FLAG-tagged β2AR, HA-tagged Gsα and EGFP were stimulated with 100 nM ISO or indicated β-blockers for 5 min. In proximity ligation assay (PLA), cells were immuno-stained with HA and β2AR antibody, nuclei were counterstained with DAPI. The green EGFP signal represents transfected cells, and red PLA signal represents Gsα and β2AR interactions. Carvedilol and alprenolol promoted Gsα recruitment to β2AR, but timolol could not. Scale bar, 10

*Figure 3 continued on next page*

*Figure 3 continued*

μm. Representative of n = 15, 16, 16, 17, 18 and 18 images respectively, three experiments. (**B–C**) HEK293 cells expressing ICUE3 biosensor were treated with 1 μM ISO or indicated β-blockers (**B**), or together with 100 μM phosphodiesterase inhibitor IBMX (**C**). (**D–E**) HEK293 cells expressing $\beta_2$AR-ICUE3 biosensor were treated with indicated concentration of ISO or β-blockers. In some cases, cells were pretreated for 30 min with the $\beta_2$AR antagonist ICI (10 μM) or the adenylate cyclase inhibitor ddA (50 μM) before adding β-blockers. Changes in ICUE3 FRET ratio (an indication of cAMP activity) were measured. Experiments were performed in the presence of 1 μM $\beta_1$AR-selective antagonist CGP20712A to block endogenous $\beta_1$AR signaling. Mock, no primary antibody; NT, no treatment; ISO, isoproterenol; TIM, timolol; ALP, alprenolol; CAR, carvedilol; ICI, ICI118551; PRO, propranolol; MET, metoprolol; 177, CGP12177, IBMX, 3-isobutyl-1-methylxanthine; ddA, 2',5'-dideoxyadenosine. Each dot in the scatter dot plot in **B–E** represents a value from an individual tested cell. Error bars denote s.e.m., *P* values are computed by one-way ANOVA followed by Tukey's test between NT (**A**) or TIM (**B–E**) and other groups.

The online version of this article includes the following source data and figure supplement(s) for figure 3:

**Source data 1.** Excel spreadsheet containing the individual numeric values of PLA dots / cell number in each raw image analyzed in *Figure 3A*, and the individual numeric values for maximum FRET responses in *Figure 3B-E*.
**Figure supplement 1.** Carvedilol-induced $\beta_2$AR phosphorylation is AC-dependent.
**Figure supplement 2.** Carvedilol- and alprenolol-induced cAMP can be abolished by $\beta_2$AR or AC inhibition.
**Figure supplement 3.** Carvedilol- and alprenolol-induced cAMP are highly restricted.

## Carvedilol augments the endogenous $\beta_2$AR-dependent PKA phosphorylation of Ca$_V$1.2 and its channel activity in hippocampal neurons

Local cAMP signals possess the potential to selectively regulate downstream effectors in receptor complexes or within the vicinity of activated receptors. In the CNS, $\beta_2$AR emerges as a prevalent postsynaptic norepinephrine effector at glutamatergic synapses, where $\beta_2$AR functionally interacts with AMPA receptor (AMPAR) and L-type Ca$^{2+}$ channel (LTCC) Ca$_V$1.2, and regulates neuronal excitability and synaptic plasticity (*Davare et al., 2001*; *Joiner et al., 2010*; *Wang et al., 2010*; *Qian et al., 2012*). CAR and ALP, but not TIM significantly increased PKA phosphorylation of S1928 and S1700 of central $\alpha_1$1.2 subunit of Ca$_V$1.2 in hippocampal neurons when both $\beta_2$AR and LTCC were endogenously expressed (*Figure 4A*, and *Figure 4—figure supplement 1A*). However, CAR and ALP failed to promote phosphorylation of the AMPAR subunit GluA1 on its PKA site serine 845 (*Figure 4B*, and *Figure 4—figure supplement 1B*). Like Ca$_V$1.2, AMPARs are associated with $\beta_2$AR, Gs, AC and PKA (*Davare et al., 2001*; *Joiner et al., 2010*; *Wang et al., 2010*; *Qian et al., 2012*). These results indicate high selectivity in targeting downstream substrates by this β-blocker-induced signaling in hippocampal neurons. Meanwhile, the CAR and ALP-induced PKA phosphorylation of LTCC were blocked by $\beta_2$AR inhibitor ICI, AC inhibitor ddA, and PKA inhibitor H89, but not CaMKII inhibitor KN93, validating the activation of $\beta_2$AR-cAMP-PKA pathway (*Figure 4C*, and *Figure 4—figure supplement 1C*). We then examined the effects of CAR on PKA-dependent activation of LTCC Ca$_V$1.2 channels using cell-attached single channel recordings in hippocampal neurons. As shown before, ISO stimulates LTCC activity (*Figure 5*) (*Shen et al., 2018*; *Davare et al., 2001*). Consistent with the phosphorylation data, CAR but not TIM significantly increased the open probability, channel availability and mean ensemble average of endogenous LTCC in rat hippocampal neurons (*Figure 5* and *Figure 5—figure supplement 1*). CAR stimulated channel activity when present in the patch pipette solution but not when applied outside the patch via bath perfusion (*Figure 5* and *Figure 5—figure supplement 1D*). Moreover, backfilling experiments with CAR found that L-type channels activity was relatively low at the beginning of the recording but then it significantly increased as the drug diffused to the pipette tip (*Figure 5F–G* and *Figure 5—figure supplement 1C*). Consistent with our hypothesis and prior studies (*Davare et al., 2001*), ISO applied outside the patch via bath perfusion was still able to stimulate LTCC activity (*Figure 5—figure supplement 1B*). These data indicate that CAR promotes spatially restricted cAMP/PKA activities for selective augmentation of LTCC activities in neurons. We further found that the activation of LTCC by CAR promoted cell death in cortical neuron cultures, and inhibition of $\beta_2$AR or LTCC counteracted carvedilol-induced neuronal toxicities (*Figure 5—figure supplement 2*).

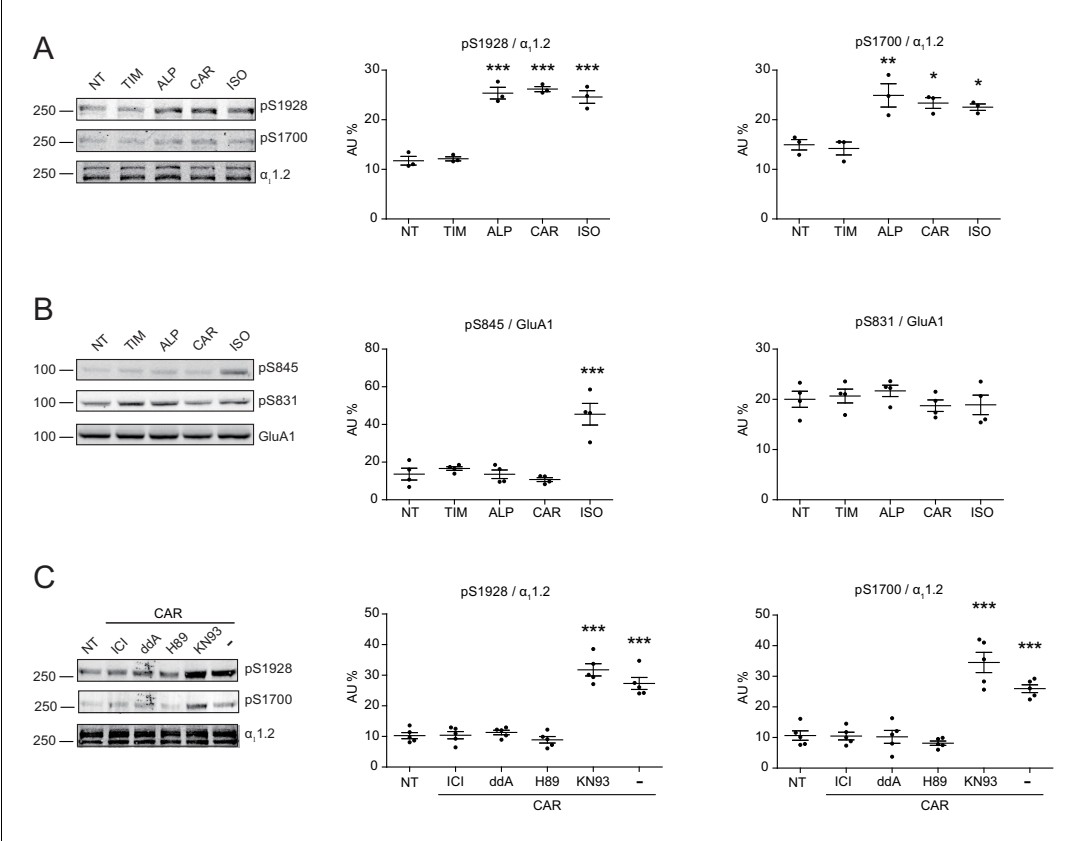

**Figure 4.** Carvedilol promotes endogenous β$_2$AR-dependent phosphorylation of LTCC α$_1$1.2 by PKA in neurons. (**A**) Rat neurons on 10–14 days in vitro (DIV) were treated for 5 min with 1 μM indicated drugs. The phosphorylation of endogenous LTCC α$_1$1.2 subunit was determined with phospho-specific antibodies, and normalized to total α$_1$1.2, n = 3. (**B**) Rat neurons on 10–14 DIV were treated for 5 min with 1 μM indicated drugs. The phosphorylation of endogenous AMPAR GluA1 subunit was determined with phospho-specific antibodies, and signals were normalized to total GluA1, n = 4. (**C**) Neurons were pretreated for 30 min with 10 μM β$_2$AR inhibitor ICI, 50 μM AC inhibitor ddA, 10 μM PKA inhibitor H89 or 10 μM CaMKII inhibitor KN93 and then stimulated with 1 μM CAR for 5 min. Carvedilol-induced LTCC phosphorylation depends on endogenous β$_2$AR, AC and PKA, but not CaMKII, n = 5. NT, no treatment; ISO, isoproterenol; TIM, timolol; ALP, alprenolol; CAR, carvedilol. Error bars denote s.e.m., *P* values are computed by one-way ANOVA followed by Tukey's test between NT and other groups.

The online version of this article includes the following source data and figure supplement(s) for figure 4:

**Source data 1.** Excel spreadsheet containing the individual numeric values of phosphorylated a$_1$1.2 or GluA1 / total a$_1$1.2 or GluA1 relative density analyzed in *Figure 4*.

**Figure supplement 1.** Uncropped blots for *Figure 4*.

## Carvedilol but not isoproterenol selectively activates a mutant β$_2$AR to augment LTCC activity in neurons

Structure-functional analyses of β$_2$AR have previously revealed distinct residues important for binding to catecholamines and β-blockers (*Strader et al., 1989*; *Liapakis et al., 2000*; *Warne et al., 2012*; *Ring et al., 2013*). We hypothesized that mutation of Ser204 and Ser207 sites within β$_2$AR binding pocket would abolish receptor hydrogen bonds with the catecholamine phenoxy moieties, thus reducing binding affinity to agonist ISO while having no effect on β-blocker binding (*Figure 6A*). Such a mutant β$_2$AR could thus be selectively activated by CAR. We co-expressed the cAMP biosensor ICUE3 together with either wild-type (WT) β$_2$AR or mutant S204A/S207A β$_2$AR in MEF cells lacking endogenous β$_1$AR and β$_2$AR (DKO) to detect receptor signaling induced by different ligands. The mutant S204A/S207A β$_2$AR induced a moderate cAMP signal at high but not low concentrations of ISO (*Figure 6B*). In contrast, after stimulation with CAR, the β$_2$AR mutant S204A/S207A promoted significant cAMP signals at nanomolar concentrations; the overall concentration response curve was similar to those induced by WT β$_2$AR (*Figure 6B*). Accordingly, the ISO-induced

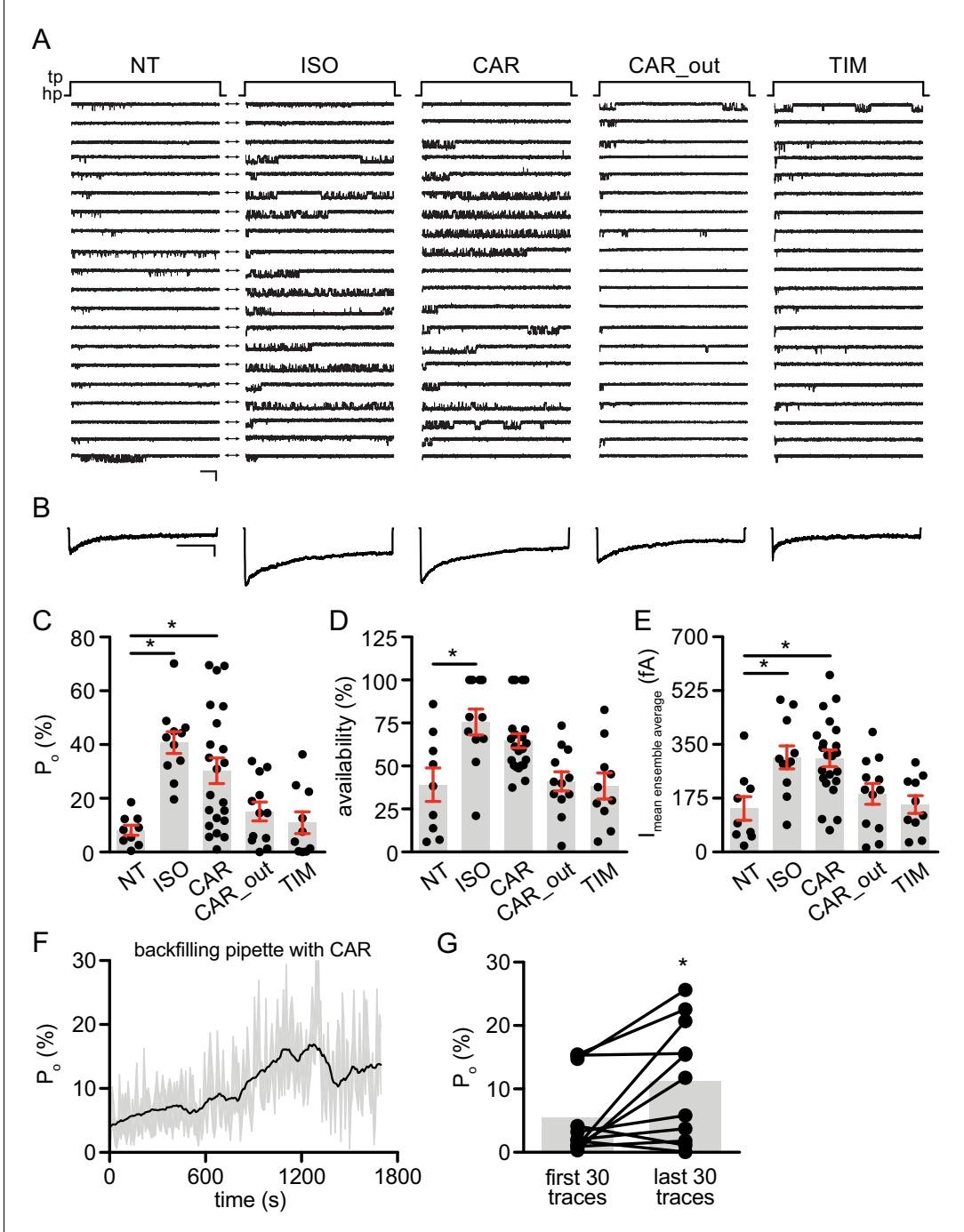

**Figure 5.** Carvedilol augments LTCC Ca$_V$1.2 channel activity in neurons. (**A**) Representative single channel recordings of LTCC Ca$_V$1.2 currents using 110 mM Ba$^{2+}$ as charge carrier in rat hippocampal neurons on 7–10 days in vitro (DIV) after depolarization from −80 (hp) to 0 mV (tp) in control patches (NT), patches containing 1 µM isoproterenol (ISO), 1 µM carvedilol (CAR) or 1 µM timolol (TIM) in the patch pipette or after addition of 1 µM CAR to the bath while the patch pipette contained a control pipette solution (CAR_out). Shown are 20 consecutive sweeps from representative experiments. Arrows throughout the figure indicate the 0-current level (closed channel). Scale bar denotes 2 pA and 200 ms. (**B**) Ensemble average currents as determined from all sweeps recorded for all the experimental conditions. Scale bar denotes 50 fA and 400 ms. (**C–E**) Mean ± s.e.m. for (**C**) P$_o$ (%), (**D**) availability (i.e. likelihood that a sweep had at least one event) (%) and (**E**) the mean ensemble average current (fA) for each experimental condition. *p<0.05 with Kruskal Wallis – Dunn's multiple comparison test. Sweep and n numbers as well as summary statistics are in **Supplementary file 1**. (**F**) Ensemble P$_o$ versus time measurements obtained with a pipette backfilled with 1 µM CAR. The solid dark line represents the mean P$_o$ over time and the gray area is the s.e.m. at each time point. The mean line was smoothed to 15 neighbors on each size with a second order polynomial smoothing in

*Figure 5 continued on next page*

*Figure 5 continued*

PRISM for representation purposes only. (G) Mean Po of the first 30 traces versus the last 30 traces obtained with a pipette backfilled with 1 μM CAR. The gray boxes highlight the mean on each group. n = 11 patches. *p<0.05 with Mann-Whitney test.

The online version of this article includes the following source data and figure supplement(s) for figure 5:

**Source data 1.** Excel spreadsheet containing the individual numeric values of Po, availability and current analyzed in *Figure 5*C-G.
**Figure supplement 1.** Over-time effect of carvedilol on LTCC of neurons recorded in the cell attached configuration without using BayK.
**Figure supplement 2.** Inhibition of $\beta_2$AR or LTCC counteracts carvedilol-induced cell death of cultured cortical neurons.

---

PKA phosphorylation of $\beta_2$AR S204A/S207A mutant was selectively abolished at nanomolar concentrations. At higher concentrations, ISO was able to induce reduced PKA phosphorylation of the $\beta_2$AR S204A/S207A mutant when compared to WT β2AR, consistent with the data of cAMP signals (*Figure 6C*, and *Figure 6—figure supplement 1A*). Meanwhile, ISO failed to induce GRK phosphorylation of $\beta_2$AR S204A/S207A mutant at different concentrations (*Figure 6C*). In comparison, CAR induced equivalent PKA phosphorylation of $\beta_2$AR WT and S204A/S207A mutant at different concentrations (*Figure 6D*, and *Figure 6—figure supplement 1B*). These data suggest that CAR, but not ISO selectively activates the S204A/S207A mutant $\beta_2$AR at nanomolar concentrations. We then tested the effects of $\beta_2$AR S204A/S207A mutant on LTCC channel activity after treatment with CAR in hippocampal neurons. In DKO neurons expressing the mutant S204A/S207A $\beta_2$AR, CAR, but not ISO (30 nM) promoted PKA phosphorylation of LTCC $\alpha_1$1.2 (*Figure 7A and B* and *Figure 7—figure supplement 1*). In agreement, CAR, but not ISO significantly increased the open probability, channel availability and mean ensemble average of LTCC (*Figure 7C–7G*). Together, CAR but not ISO selectively activates the S204A/S207A mutant $\beta_2$AR at low concentrations and increases channel opening probabilities.

## Discussion

In a classic view, agonist stimulation promotes both PKA and GRK phosphorylation of activated GPCRs. Although previous studies have reported that some β-blockers promote βAR-Gs coupling and thus might display partial agonism, this phenomenon is only observed at high concentrations and in vitro with reconstituted systems (*Yao et al., 2009*; *DeVree et al., 2016*; *Gregorio et al., 2017*). In this study, using a combination of highly sensitive tools such as engineered FRET-based cAMP sensors and single channel recording together with detection by phospho-specific antibodies, we show for the first time that β-blockers such as CAR and ALP can promote receptor-Gs coupling at nanomolar concentrations in living cells, which is clinically relevant in contrast to superphysiological concentrations in previous studies. In detail, as low as 1 nM alprenolol as well as 1 nM carvedilol induce 20–40% of maximal effects (as obtained with 1 μM isoproterenol) with respect to phosphorylation of $\beta_2$AR by PKA and to cAMP production detected by the ICUE3 sensor coupled to $\beta_2$AR. Unlike agonists, activation of $\beta_2$AR by β-blockers selectively transduce G protein/cAMP/PKA signaling but not GRK signaling. More importantly, the $\beta_2$AR-induced cAMP signal is highly spatially restricted to the local domain of activated $\beta_2$AR, which selectively promotes activation of receptor-associated LTCC but not receptor-associated AMPAR, two downstream ion channels essential for adrenergic regulation of neuronal excitability in hippocampal neurons. The differential signaling by carvedilol with respect to LTCC and AMPAR is especially remarkable because both channels form complexes with $\beta_2$AR that are localized within dendritic spines. Moreover, we have engineered a mutant $\beta_2$AR that is selectively activated by β-blockers but not by catecholamines at low concentration. Our study defines CAR and ALP as Gs-biased partial agonists of βAR for highly spatially restricted cAMP/PKA signaling to $Ca_V$1.2 in neurons. The study exemplifies a unique mechanism by which β-blockers shape the compartmentalization of βAR signaling and a highly restrictive distribution of ligand-induced activation of GPCR targeting a specific downstream effector.

PKA-mediated phosphorylation is thought to play critical roles in heterologous desensitization of GPCRs and in receptor switching from Gs to Gi coupling (*Daaka et al., 1997*; *Zamah et al., 2002*), whereas GRK-mediated phosphorylation is implicated in β-arrestin recruitment and β-arrestin-dependent ERK activation (*Luttrell et al., 1999*; *Pierce et al., 2000*; *Kim et al., 2005*; *Ren et al., 2005*; *Zidar et al., 2009*; *Choi et al., 2018*). We have recently characterized that PKA and GRKs phosphorylate distinct subpopulations of $\beta_2$AR in a single fibroblast or neuron (*Shen et al., 2018*). While GRK

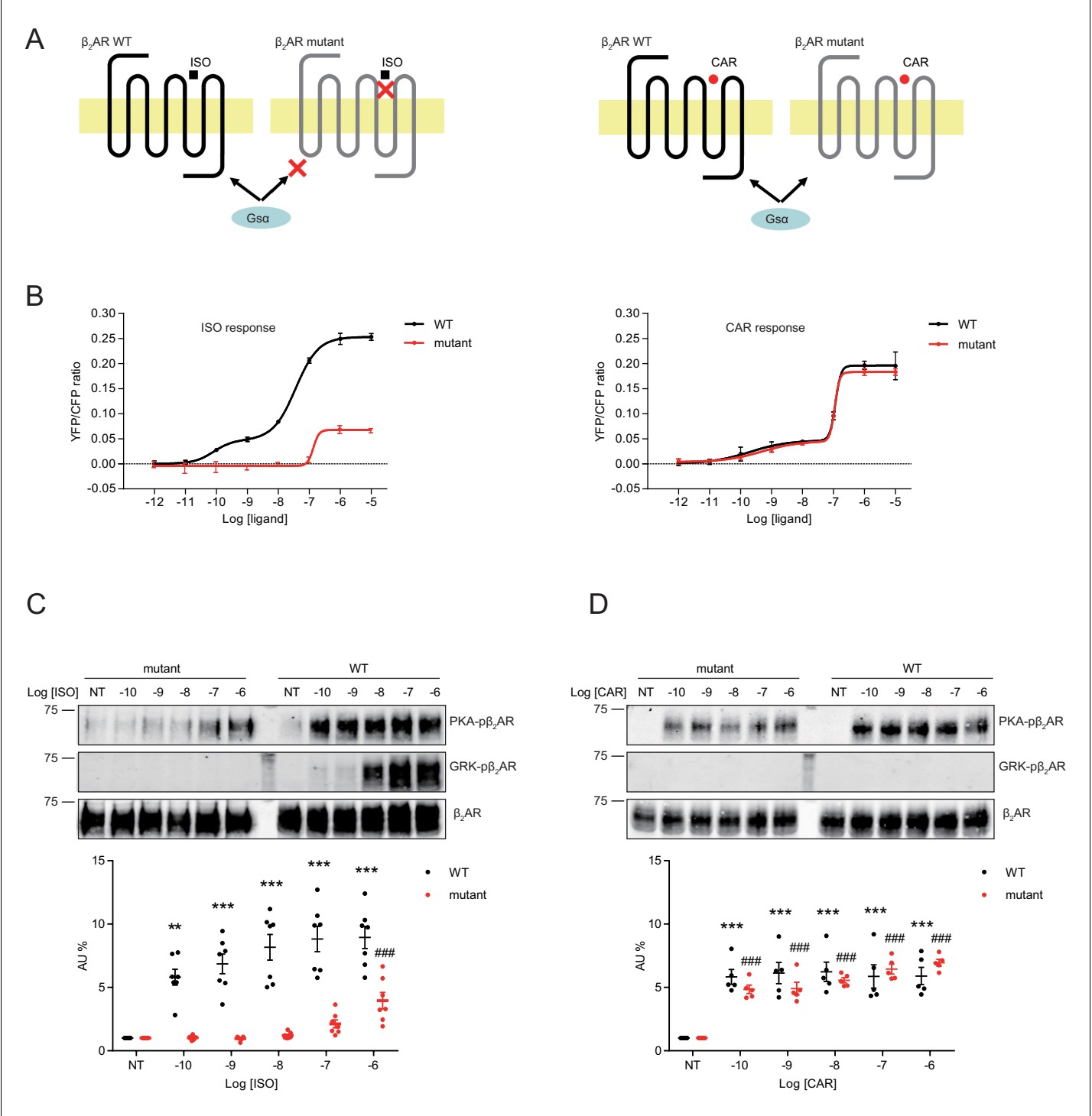

**Figure 6.** A mutant $\beta_2$AR is selectively activated by carvedilol but not isoproterenol. (**A**) Schematic of an engineered $\beta_2$AR with S204/207A double serine mutations that loses high affinity binding to ISO but not CAR at nanomolar range. (**B**) cAMP biosensor ICUE3 and $\beta_2$AR wild-type (WT) or mutant were co-expressed in MEF cells lacking both $\beta_1$AR and $\beta_2$AR. Changes of cAMP FRET ratio by increasing concentrations of ISO or CAR were measured. n = 5–29 cells. (**C–D**) HEK293 cells stably expressing FLAG-tagged $\beta_2$AR WT or mutant were stimulated for 5 min with increasing concentrations of ISO (**C**), n = 7 or CAR (**D**), n = 5. The phosphorylation of $\beta_2$AR on its PKA and GRK sites were determined with phospho-specific antibodies, and signals were normalized to total $\beta_2$AR detected with anti-FLAG antibody. Experiments were performed in the presence of 1 µM $\beta_1$AR-selective antagonist CGP20712A to block endogenous $\beta_1$AR signaling. NT, no treatment; ISO, isoproterenol; CAR, carvedilol. Error bars denote s.e.m., *P* values are computed by one-way ANOVA followed by Tukey's test between NT and other concentrations.

The online version of this article includes the following source data and figure supplement(s) for figure 6:

*Figure 6 continued on next page*

*Figure 6 continued*

**Source data 1.** Excel spreadsheet containing the individual numeric values for maximum FRET responses in *Figure 6B*, and the individual numeric values of phosphorylated $\beta_2AR$ / total $\beta_2AR$ relative density analyzed in *Figure 6C-D*.
**Figure supplement 1.** Uncropped blots for *Figure 6C and D*.

phosphorylation of $\beta_2AR$ is only observed at high concentrations of agonists, PKA phosphorylation can be induced with minimal doses of agonist (*Shen et al., 2018*; *Tran et al., 2004*; *Tran et al., 2007*; *Liu et al., 2009*). Here, our data show CAR does not promote GRK phosphorylation at low concentrations and induces a slow and minimal GRK effect at high concentrations when compared to those induced by ISO. The CAR-induced GRK effects are minimally related to the PKA effects. Previously, CAR has been recognized as a biased $\beta$-blocker that preferentially activates $\beta$-arrestin/ ERK pathways (*Wisler et al., 2007*; *Kim et al., 2008*). Despite the prominent role of GRK phosphorylation in full agonist ISO-induced $\beta_2AR$-$\beta$-arrestin/ERK signaling, our data clearly indicate that GRK phosphorylation of $\beta_2AR$ is not necessary for CAR-induced activation of ERK, consistent with a recent study showing a distinct general mechanism of $\beta$-arrestin activation that does not require the GRK-phosphorylated tail of different GPCRs (*Eichel et al., 2018*). Meanwhile, other studies show that in the absence of all G proteins, GPCRs fail to transduce $\beta$-arrestin/ERK signaling (*Grundmann et al., 2018*). These data indicate the necessity of G proteins in GPCR-induced arrestin activation. In our study, we observed a concentration-dependent correlation between PKA phosphorylation of $\beta_2AR$ with ERK activity induced by $\beta$-blockers, suggesting the potential role of Gs and PKA in CAR-induced $\beta_2AR$-$\beta$-arrestin/ERK signaling are overlooked. In comparison, Gi is not required for CAR-induced $\beta_2AR$/$\beta$-arrestin signaling even though CAR induces Gi recruitment to $\beta_1AR$ for transducing $\beta_1AR$/$\beta$-arrestin signaling (*Wang et al., 2017*). Moreover, our results are also in line with a recent report that activation of $\beta_2AR$ with as low as femtomolar concentrations of ligands causes sustained ERK signaling (*Civciristov et al., 2018*), further support a PKA but GRK-dependent mechanism in GPCR-induced ERK activation. Future studies will help us understand how ligand-induced GPCRs utilize distinct mechanisms in activating $\beta$-arrestin/ERK pathway.

Engineered GPCRs have been widely applied in investigating structural and biological processes and behaviors by precisely controlling specific GPCR signaling branches (*Lee et al., 2014*). Previous mutagenesis studies have shown that $\beta_2AR$ with S204/207A mutation loses binding to adrenaline but still binds with several $\beta$-blockers including ALP (*Liapakis et al., 2000*). Based on this and recent advances in $\beta$AR structures with agonists and $\beta$-blockers (*Warne et al., 2012*; *Ring et al., 2013*), we have generated a S204/207A mutant that bestow $\beta_2AR$ with the ability to be selectively activated by $\beta$-blockers such as CAR and to transduce cAMP/PKA signaling. At nanomolar concentrations, while ISO fails to stimulate PKA phosphorylation of the S204/207A mutant $\beta_2AR$, the mutant receptor still retains CAR-induced stimulation of PKA-phosphorylation of the receptor. The CAR-induced activation of mutant $\beta_2AR$ triggers the $\beta_2AR$/Gs/cAMP/PKA signaling pathway and selectively targets downstream effectors in primary hippocampal neurons. Interestingly, the S204/207A $\beta$2AR mutant is not only refractory to its agonists but also completely lost both ISO- and CAR-induced GRK-phosphorylation of $\beta_2AR$. Further studies comparing this mutant with previous reported $\beta_2AR$-TYY and Y219A mutants that lack Gs and GRKs coupling, respectively (*Choi et al., 2018*; *Shenoy et al., 2006*), will facilitate the analysis of the physiological relevance of Gs/cAMP/PKA-dependent and GRK-dependent signaling pathways and enable researchers to explore $\beta$-arrestin/ERK pathway devoid of individual signaling branches.

$\beta$-blockers are a standard clinical treatment in a broad range of diseases. Many $\beta$-blockers possess intrinsic sympathomimetic activities (*Bakris, 2009*; *Gorre and Vandekerckhove, 2010*), which are problematic due to the side effects through stimulation of $\beta$ARs (*Bakris, 2009*; *Gorre and Vandekerckhove, 2010*), a feature that limits the clinical utility of the drugs. Here, we show that $\beta$-blockers promote activation of $\beta_2AR$ by recruiting Gs that selectively transduces cAMP/PKA signal but not GRK signal. Meanwhile, binding of $\beta$-blockers to $\beta_1AR$ has been shown to enhance cAMP levels locally by dissociating a $\beta_1AR$-PDE4 complex, thereby reducing the local cAMP-hydrolytic activity (*Richter et al., 2013*), $\beta_1AR$ and $\beta_2AR$ thus could utilize different mechanisms for $\beta$-blocker-induced signaling. Another interesting observation is that the $\beta$-blocker-induced $\beta_2AR$-cAMP signal is sufficient to promote PKA phosphorylation of both $\beta_2AR$ and the receptor-associated $Ca_V1.2$ of LTCC,

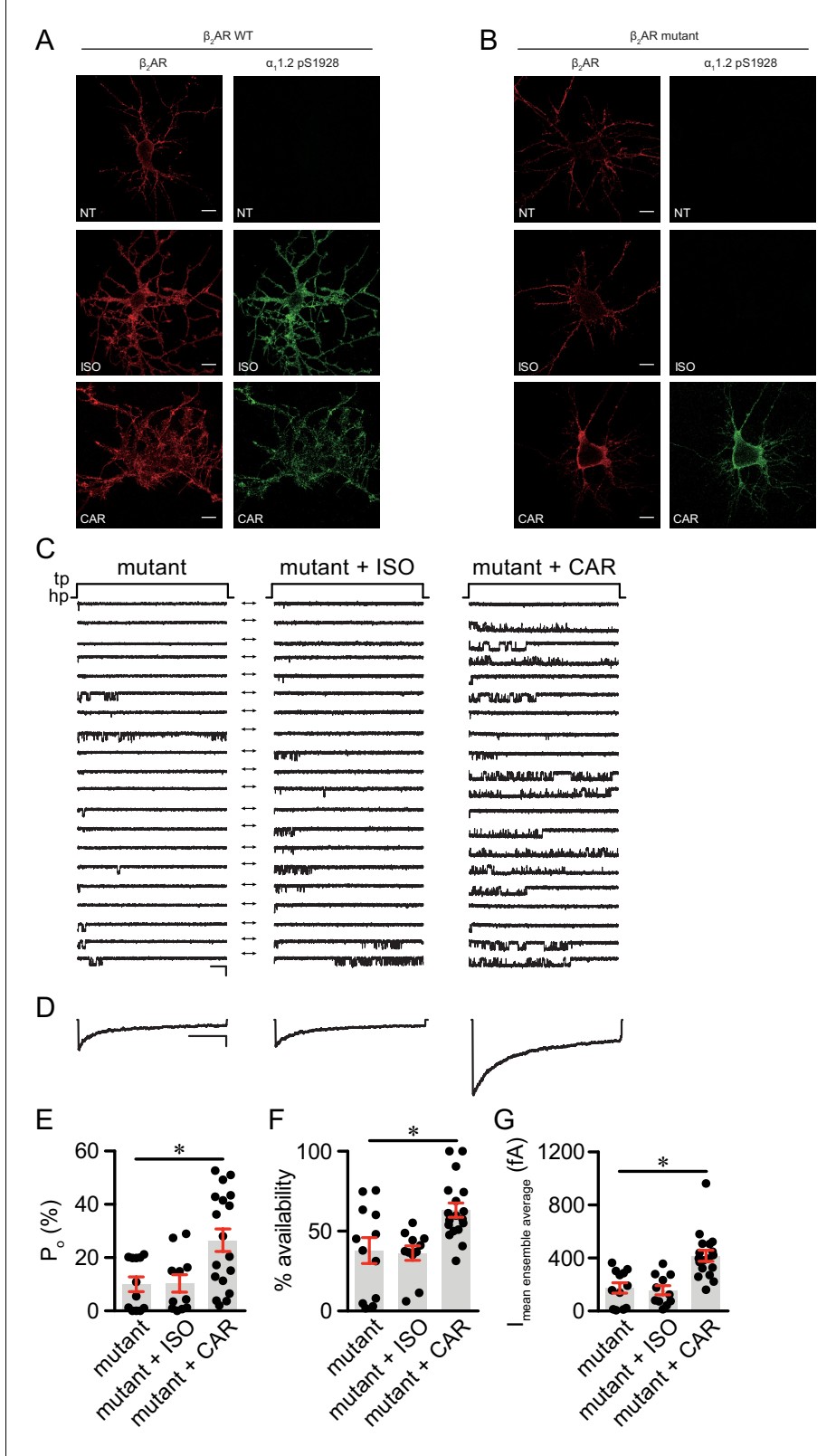

**Figure 7.** The $\beta_2$AR mutant selectively supports carvedilol-induced augmentation of LTCC activity in neurons. (A–B) $\beta_1$AR/$\beta_2$AR double knockout (DKO) mouse hippocampal neurons on 7–10 days in vitro (DIV) were cotransfected with FLAG-tagged $\beta_2$AR WT (A) or mutant (B) and HA-tagged LTCC $\alpha_1$1.2 subunit, 24 hr later cells were either mock treated (NT), or treated for 5 min with 10 nM isoproterenol (ISO) or carvedilol (CAR), fixed and labeled with

*Figure 7 continued on next page*

*Figure 7 continued*

anti-FLAG and a phospho-specific antibody for S1928 phosphorylated $\alpha_1$1.2. Confocal images show mutant $\beta_2$AR losts the ability of promoting LTCC phosphorylation upon ISO stimulation but remained the ability upon CAR stimulation in neurons. Scale bar, 10 μm. Representative of 6 images for each condition, three experiments. (**C**) Representative single channel recordings of LTCC $Ca_V$1.2 currents using 110 mM $Ba^{2+}$ as charge carrier in DKO neurons on 7–10.days DIV expressing mutant $\beta_2$AR after depolarization from −80 to 0 mV in in control patches (mutant) and patches containing 1 μM isoproterenol (ISO) or 1 μM carvedilol (CAR) in the patch pipette. Shown are 20 consecutive sweeps from representative experiments. Arrows throughout the figure indicate the 0-current level (closed channel). Scale bar denotes 2 pA and 200 ms. (**D**) Ensemble average currents as determined from all sweeps recorded for all the experimental conditions. Scale bar denotes 50 fA and 400 ms. (**E–G**) Mean ± s.e.m. for (**E**) $P_o$ (%), (**F**) availability (i.e. likelihood that a sweep had at least one event) (%) and (**G**) the mean ensemble average current (fA) for each experimental condition. *p<0.05 with Kruskal Wallis – Dunn's multiple comparison test. Sweep and n numbers as well as summary statistics are in *Supplementary file 2*.

The online version of this article includes the following source data and figure supplement(s) for figure 7:

**Source data 1.** Excel spreadsheet containing the individual numeric values of Po, availability and current analyzed in *Figure 7E-G*.

**Figure supplement 1.** The mutant $\beta_2$AR is selectively activated by carvedilol and promotes LTCC phosphorylation in neurons.

---

but not another substrate, the AMPAR GluA1 subunit. Both LTCC and AMPAR are shown to associate with the $\beta_2$AR in hippocampal neurons (*Davare et al., 2001*; *Joiner et al., 2010*; *Wang et al., 2010*; *Qian et al., 2012*). Therefore, the preference of one local membrane target over another local target indicates a highly restricted nature of the cAMP-PKA activities, potentially dependent on the recently identified distinct subpopulations of $\beta_2$AR and associated signaling molecules in the neurons (*Shen et al., 2018*). Nevertheless, the PKA phosphorylation leads to augmentation of LTCC activity, potentially contributing to the neuronal toxicities. Therefore, activation of GPCR at low ligand concentrations should be taken into consideration when designing and screening new therapeutic drugs.

## Materials and methods

### Key resources table

| Reagent type (species) or resource | Designation | Source or reference | Identifiers | Additional information |
|---|---|---|---|---|
| Strain (*Mus musculus*) | $\beta_1$AR/$\beta_2$AR double knockout | Jackson Laboratories | Stock # 003810 | |
| Strain (*Rattus norvegicus*) | Sprague Dawley | Charles River Laboratories | | |
| Cell line (*Homo sapiens*) | HEK293/$\beta_2$AR-WT | *De Arcangelis et al., 2009* | | HEK293 cells stably expressing FLAG-$\beta_2$AR |
| Cell line (*Homo sapiens*) | HEK293/$\beta_2$AR-S204/207A | This paper | | HEK293 cells stably expressing FLAG-$\beta_2$AR-S204/207A |
| Antibody | Phospho-$\beta_2$AR (Ser261/262) (mouse monoclonal) | Dr. Richard Clark (UT Huston) | Clone 2G3 | IF (1 μg/ml), WB (1:1000) |
| Antibody | Phospho-$\beta_2$AR (Ser355/356) (mouse monoclonal) | Dr. Richard Clark (UT Huston) | Clone 10A5 | WB (1:1000) |
| Antibody | $\beta_2$AR (rabbit polyclonal) | Santa Cruz Biotechnology | sc-570 RRID:AB_2225412 | PLA (1:100), WB (1:1000) |
| Antibody | Phospho-$\beta_2$AR (Ser355/356) (rabbit polyclonal) | Santa Cruz Biotechnology | sc-16719R RRID:AB_781609 | WB (1:1000) |

*Continued on next page*

*Continued*

| Reagent type (species) or resource | Designation | Source or reference | Identifiers | Additional information |
|---|---|---|---|---|
| Antibody | $\alpha_1$1.2 (rabbit polyclonal) | *Patriarchi et al., 2016* | FP1 | WB (1:1000) |
| Antibody | Phospho-$\alpha_1$1.2 (Ser1928) (rabbit polyclonal) | *Patriarchi et al., 2016* | CH3P | IF (1 µg/ml), WB (1:1000) |
| Antibody | Phospho-$\alpha_1$1.2 (Ser1700) (rabbit polyclonal) | *Patriarchi et al., 2016*, Originally from Dr. William Catterall (U of Washington) | | WB (1:1000) |
| Antibody | GluA1 (rabbit polyclonal) | *Patriarchi et al., 2016* | | WB (1:1000) |
| Antibody | Phospho-GluA1 (Ser831) (rabbit polyclonal) | *Patriarchi et al., 2016* | | WB (1:1000) |
| Antibody | Phospho-GluA1 (Ser845) (rabbit polyclonal) | *Patriarchi et al., 2016* | | WB (1:1000) |
| Antibody | FLAG-M1 | Sigma-Aldrich | F3040 RRID:AB_439712 | IF (1 µg/ml), WB (1:1000) |
| Antibody | HA | Covance | MMS-101R RRID:AB_291262 | PLA (1:1000) |
| Recombinant DNA reagent | $\beta_2$AR-mutant | This paper | | FLAG-tagged human $\beta_2$AR with S204/207A double mutations |
| Recombinant DNA reagent | $\beta_2$AR-ICUE3 | This paper | | ICUE3 fused to the C-terminal of human $\beta_2$AR |
| Commercial assay or kit | Duolink in situ detection reagents | Sigma-Aldrich | DUO92007 | PLA |
| Software, algorithm | pCLAMP10 | Molecular Devices | | electrophysiology |
| Software, algorithm | MetaFluor | Molecular Devices | | FRET |

## Animals

$\beta_1$AR/$\beta_2$AR double knockout (DKO) mouse were obtained from Jackson Laboratories to produce P0-P1 postnatal DKO pups, SD pregnant rats were obtained from Charles River Laboratories to provide E17-E19 embryonic rats. All of the animals were handled according to approved institutional animal care and use committee (IACUC) protocols (#20234 and #20673) of the University of California at Davis and in accordance with the NIH guidelines.

## Plasmids

DNA constructs expressing FLAG-tagged human $\beta_2$AR (FLAG-$\beta_2$AR) and HA-tagged rat L-type calcium channel (LTCC) $\alpha_1$1.2 were described before (*Shen et al., 2018*). FLAG-tagged human $\beta_2$AR with S204/207A double mutations (FLAG-mutant) was generated by Gibson assembly method (Thermo Fisher) using FLAG-$\beta_2$AR and synthetic gBlocks with the double mutations as templates (Integrated DNA Technologies). FRET biosensor ICUE3, CAAX-ICUE3 and LYN-ICUE3 were described before (*DiPilato and Zhang, 2009*). To make the $\beta_2$AR-ICUE3 fusion biosensor, ICUE3 was fused to the C-terminal of FLAG-$\beta_2$AR with Gly-Ser linker. HA-Gs$\alpha$ was made by replacing CFP with HA tag, using Gs$\alpha$-CFP as template (a gift from Dr. Catherine Berlot, Addgene plasmid # 55793).

## Antibodies and chemicals

Mouse monoclonal antibodies against $\beta_2AR$ at serine 261/262 (clone 2G3) and at serine 355/356 (clone 10A5) were kindly provided by Dr. Richard Clark (UT Huston). Polyclonal antibodies against $\beta_2AR$ (sc-570) and phosphorylated $\beta_2AR$ at serine 355/356 (sc-16719R) were purchased from Santa Cruz Biotechnology. Polyclonal antibodies against $\alpha_11.2$ residues 754–901 for total $\alpha_11.2$ (FP1), residues 1923–1935 for phosphorylated serine 1928 site (LGRRApSFHLECLK, pS1928) and residues 1694–1709 for phosphorylated serine 1700 site (EIRRAIpSGDLTAEEEL, pS1700) were described before (*Patriarchi et al., 2016*). Polyclonal antibodies against GluA1 residues 894–907 for total GluA1, residues 826–837 for phosphorylated serine 831 site (LIPQQpSINEAIK, pS831) and residues 840–851 for phosphorylated serine 845 site (TLPRNpSGAGASK, pS845) were described before (*Patriarchi et al., 2016*). Other antibodies used in the experiments include: anti-FLAG (F3040, Sigma), anti-HA (MMS-101R, Covance), Alexa fluor 488 conjugated goat anti-rabbit IgG and Alexa fluor 594 conjugated goat anti-mouse IgG (A-11034 and A-11032, Thermo Fisher), DyLight 680 conjugated goat anti-mouse IgG and anti-rabbit IgG (35518 and 35568, Thermo Fisher), IRDye 800CW conjugated goat anti-mouse IgG and anti-rabbit IgG (926–32210 and 926–32211, Li-cor).

Isoproterenol (I2760), timolol (T6394), alprenolol (A8676), propranolol (P0884), metoprolol (M5391), CGP12177A (C125), CGP20712A (C231), ICI118551 (I127), 3-isobutyl-1-methylxanthine (I5879) and 2',5'-dideoxyadenosine (D7408) were purchased from Sigma. Carvedilol (15418) was from Cayman Chemical, H89 (H-5239) was from LC Labs, pertussis toxin (179B) was from List Labs.

## Cell culture and transfection

Human embryonic kidney HEK293 cells were from American Type Culture Collection (ATCC) and were maintained in Dulbecco's modified Eagle medium (DMEM, Corning) supplemented with 10% fetal bovine serum (FBS, Sigma). HEK293 cells stably expressing FLAG-$\beta_2AR$ was from previous study (*De Arcangelis et al., 2009*). HEK293 cells stably expressing FLAG-mutant $\beta_2AR$ was generated in this study. Briefly, cells transfected with $\beta_2AR$-mutant were selected by G418 resistance (Corning) and cell clones were obtained by limiting serial dilution, monoclonal cells were analyzed by western blots and the one with comparable $\beta_2AR$ expression to FLAG-$\beta_2AR$ stable cells was chosen.

Mouse embryonic fibroblasts (MEFs) from $\beta_1AR/\beta_2AR$ double knockout (DKO) mouse were described in previous study (*Cervantes et al., 2010*) and were maintained in DMEM supplemented with 10% FBS. Primary mouse hippocampal neurons were isolated and cultured from P0-P1 early postnatal DKO mouse pups, and primary rat hippocampal neurons were prepared from E17-E19 embryonic rats using methods described previously (*Chen et al., 2008*; *Beaudoin et al., 2012*). Briefly, dissected hippocampi were dissociated by 0.25% trypsin (Corning) and trituration. Neurons were plated on poly-D-lysine-coated (Sigma) glass coverslips in 24-well plate for imaging and in 6-well plate for biochemistry at a cell density of 50,000/well and 1 million/well, respectively. Neurons were cultured in Neurobasal medium supplemented with GlutaMax and B-27 (Thermo Fisher).

HEK293 cells were transfected with plasmids using polyethylenimine according to manufacturer's instructions (Sigma). Neurons were transfected by the $Ca^{2+}$-phosphate method (*Jiang and Chen, 2006*). Briefly, cultured neurons on 6–10 DIV were switched to pre-warmed Eagle's minimum essential medium (EMEM, Thermo Fisher) supplemented with GlutaMax 1 hr before transfection, conditioned media were saved. DNA precipitates were prepared by 2x HBS (pH 6.96) and 2 M $CaCl_2$. After incubation with DNA precipitates for 1 hr, neurons were incubated in 10% $CO_2$ pre-equilibrium EMEM for 20 min, then replaced with conditioned medium and cultured in 5% $CO_2$ incubator until use.

## Confocal microscopy imaging

Rat hippocampal neurons were transfected with FLAG-$\beta_2AR$ on 10 DIV, treated for 5 min with 10 nM or 1 $\mu$M indicated drugs on 12 DIV. Mouse DKO hippocampal neurons were transfected with FLAG-$\beta_2AR$ or FLAG-mutant and HA-$\alpha_11.2$ at 1:1 ratio on 6–8 DIV, and stimulated with indicated drugs and times 24 hr after transfection. Treated cells were fixed, permeabilized, and co-stained with indicated antibodies with a final concentration of 1 $\mu$g/ml for each antibody, which were revealed by a 1:1000 dilution of Alexa fluor 488 conjugated goat anti-rabbit IgG or Alexa fluor 594 conjugated goat anti-mouse IgG, respectively. Fluorescence images were taken by Zeiss LSM 700 confocal microscope with a 63×/1.4 numerical aperture oil-immersion lens.

## Proximity ligation assay

HEK293 cells growing on poly-D-lysine coated coverslips were transfected with FLAG-$\beta_2$AR or FLAG-mutant, HA-Gs$\alpha$ and pEYFP-N1 at 8:1:1 ratio. 24 hr after transfection, cells were serum-starved 2 hr, treated 100 nM indicated drugs for 5 min. Following stimulation, cells were fixed, permeabilized, and co-stained with anti-$\beta_2$AR antibody (1:100 dilution) from rabbit in conjunction with anti-HA antibody (1:1000 dilution) from mouse. The proximity ligation reaction was performed according to the manufacturer's protocol using the Duolink in situ detection orange reagents (Sigma). Images were recorded with Zeiss LSM 700 confocal microscope with a 63×/1.4 numerical aperture oil-immersion lens. To quantify the PLA signals, the number of red fluorescent objects in each image was quantified using the Squassh plug-in for ImageJ software (*Rizk et al., 2014*), and divided by the number of transfected cells.

## Fluorescence resonance energy transfer (FRET) measurement

FRET measurement was performed as previously described (*De Arcangelis et al., 2009*). Briefly, HEK 293 cells were transfected with ICUE3 or $\beta_2$AR-ICUE3, DKO MEFs were co-transfected with ICUE3 and FLAG-$\beta_2$AR or FLAG-mutant. Cells were imaged on a Zeiss Axiovert 200M microscope with a 40×/1.3 numerical aperture oil-immersion lens and a cooled CCD camera. Dual emission ratio imaging was acquired with a 420DF20 excitation filter, a 450DRLP diachronic mirror, and two emission filters (475DF40 for cyan and 535DF25 for yellow). The acquisition was set with 0.2 s exposure in both channels and 20 s elapses. Images in both channels were subjected to background subtraction, and ratios of yellow-to-cyan were calculated at different time points.

## Western blot

HEK293 cells stably expressing FLAG-$\beta_2$AR or FLAG-mutant were serum-starved for 2 hr and treated with indicated drugs and times, then harvested by lysis buffer (10 mM Tris pH 7.4, 1% NP40, 150 mM NaCl, 2 mM EDTA) with protease and phosphatase inhibitor cocktail. Rat hippocampal neurons on 10–14 DIV were treated with indicated drugs and times, then harvested by lysis buffer (10 mM Tris pH 7.4, 1% TX-100, 150 mM NaCl, 5 mM EGTA, 10 mM EDTA, 10% glycerol) with protease and phosphatase inhibitor cocktail. Protein samples were analyzed by western blot using antibodies as indicated at a 1:1000 dilution and signals were detected by Odyssey scanner (Li-cor).

## Cell-attached patch clamp electrophysiology

Primary rat and mouse hippocampal neurons were used on 7–10 DIV. Cell-attached patch clamp recordings were performed on an Olympus IX70 inverted microscope in a 15 mm culture coverslip at room temperature (22–25˚C). Signals were recorded at 10 kHz and low-pass filtered at 2 kHz with an Axopatch 200B amplifier and digitized with a Digidata 1440 (Molecular Devices). Recording pipettes were pulled from borosilicate capillary glass (0.86 OD) with a Flaming micropipette puller (Model P-97, Sutter Instruments) and polished (polisher from World Precision Instruments). Pipette resistances were strictly maintained between 6–7 M$\Omega$ to ameliorate variations in number of channels in the patch pipette. The patch transmembrane potential was zeroed by perfusing cells with a high K$^+$ extracellular solution containing (in mM) 145 KCl, 10 NaCl, and 10 HEPES, pH 7.4 (NaOH). The pipette solution contained (in mM) 20 tetraethylammonium chloride (TEA-Cl), 110 BaCl$_2$ (as charge carrier), and 10 HEPES, pH 7.3 (TEA-OH). This pipette solution was supplemented with 1 $\mu$M $\omega$-conotoxin GVIA and 1 $\mu$M $\omega$-conotoxin MCVIIC to block N and P/Q-type Ca$^{2+}$ channels, respectively, and (S)-(-)-BayK-8644 (500 nM) was included in the pipette solution to promote longer open times and resolve channel openings as previously performed by our group and others (*Shen et al., 2018*; *Davare et al., 2001*; *Wang et al., 2001*; *Qian et al., 2017*; *Hess et al., 1986*; *Schuhmann et al., 1997*; *Costantin et al., 1998*; *Yue and Marban, 1990*; *Dzhura and Neely, 2003*; *Navedo et al., 2005*). In a subset of experiments, BayK was left out of the pipette solution. Note that ISO and CAR had similar effects on channel activity whether BayK was included or not in the pipette solution. To examine the effects of $\beta$-adrenergic stimulation on the L-type Ca$_V$1.2 single-channel activity, 1 $\mu$M isoproterenol was added to the pipette solution in independent experiments. Note that we have previously used the L-type Ca$_V$1.2 channel blocker nifedipine (1 $\mu$M) to confirm the recording of L-type Ca$_V$1.2 currents under control conditions and in the presence of isoproterenol (*Patriarchi et al., 2016*). Single-channel activity was recorded during a single pulse protocol (2 s)

from a holding potential of −80 mV to 0 mV every 5 s. An average of >50 sweeps were collected with each recording file under all experimental conditions. The half-amplitude event-detection algorithm of pClamp10 was used to measure overall single-channel L-type $Ca_V1.2$ activity as nPo, where n is the number of channels in the patch and Po is the open probability. Because the variability of nPo can be a critical element to interpret single channel data due to overstating open probability based on a high n number, we corrected this parameter by the number of channels (n) describing channel open probability and availability as well as calculating the mean ensemble average current. Data were pooled for each condition and analyzed with GraphPad Prism software.

## Statistical analysis

Data were analyzed using GraphPad Prism software and expressed as mean ± s.e.m. Differences between two groups were assessed by appropriate two-tailed unpaired Student's t-test or nonparametric Mann-Whitney test. Differences among three or more groups were assessed by One-way ANOVA with Tukey's post hoc test or the Kruskal-Wallis test with Dunn's post hoc test. $p < 0.05$ was considered statistically significant (denoted by * or # in figures).

## Data availability

All data generated or analyzed during this study are included in the manuscript and supporting files. Source data files have been provided for all main figures.

## Acknowledgements

This work was supported by NIH grant GM129376 and VA Merit grant BX002900 to YKX, and NIH grants HL098200, HL121059 and HL149127 to MFN. AS and QS were recipients of AHA postdoctoral fellowship. YKX is an established AHA investigator.

## Additional information

### Funding

| Funder | Grant reference number | Author |
| --- | --- | --- |
| National Institutes of Health | GM129376 | Yang K Xiang |
| U.S. Department of Veterans Affairs | BX002900 | Yang K Xiang |
| National Institutes of Health | HL098200 | Manuel F Navedo |
| National Institutes of Health | HL121059 | Manuel F Navedo |
| National Institutes of Health | HL149127 | Manuel F Navedo |
| American Heart Association | Postdoctoral fellowship | Ao Shen Qian Shi |
| American Heart Association | | Yang K Xiang |

The funders had no role in study design, data collection and interpretation, or the decision to submit the work for publication.

### Author contributions

Ao Shen, Conceptualization, Data curation, Formal analysis, Writing—original draft, Writing—review and editing; Dana Chen, Manpreet Kaur, Peter Bartels, Bing Xu, Qian Shi, Joseph M Martinez, Madeline Nieves-Cintron, Data curation; Kwun-nok Mimi Man, Resources; Johannes W Hell, Supervision, Writing—review and editing; Manuel F Navedo, Data curation, Formal analysis, Writing—review and editing; Xi-Yong Yu, Supervision; Yang K Xiang, Conceptualization, Formal analysis, Supervision, Funding acquisition, Writing—original draft, Writing—review and editing

## Author ORCIDs

Ao Shen (iD) https://orcid.org/0000-0002-1559-3895
Johannes W Hell (iD) http://orcid.org/0000-0001-7960-7531
Manuel F Navedo (iD) http://orcid.org/0000-0001-6864-6594
Yang K Xiang (iD) https://orcid.org/0000-0003-1786-9143

## Ethics

Animal experimentation: This study was performed in strict accordance with the recommendations in the Guide for the Care and Use of Laboratory Animals of the National Institutes of Health. All of the animals were handled according to approved institutional animal care and use committee (IACUC) protocols (#20234) of the University of California at Davis. Every effort was made to minimize suffering.

## Decision letter and Author response

Decision letter https://doi.org/10.7554/eLife.49464.sa1
Author response https://doi.org/10.7554/eLife.49464.sa2

## Additional files

### Supplementary files

• Supplementary file 1. Biophysical properties of L-type $Ca^{2+}$ currents in the neurons recorded in *Figure 5A–5E*.Values are mean ± SEM. *$p < 0.05$ with Kruskal Wallis – Dunn's multiple comparison test.

• Supplementary file 2. Biophysical properties of L-type $Ca^{2+}$ currents in the neurons recorded in *Figure 7C–7G*.Values are mean ± SEM. *$p < 0.05$ with Kruskal Wallis – Dunn's multiple comparison test.

• Transparent reporting form

### Data availability

All data generated or analyzed during this study are included in the manuscript and supporting files. Source data files have been provided for all main figures.

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
