## [Decision Letter]

Thank you for submitting your article "β-blockers augment LTCC activity by targeting spatially restricted β2AR signaling in neurons" for consideration by *eLife*. Your article has been reviewed by three peer reviewers, and the evaluation has been overseen by a Reviewing Editor and Olga Boudker as the Senior Editor. The following individual involved in review of your submission has agreed to reveal their identity: Ruiping Xiao (Reviewer #3).

The reviewers have discussed the reviews with one another and the Reviewing Editor has drafted this decision to help you prepare a revised submission.

Summary:

The authors expand on their recent studies exploring the local regulation of L-type Ca channels (LTCC) in neurons by β-adrenergic signaling. Previously, they found that cAMP-PKA dependent phosphorylation of beta2 adrenergic receptors controls ion channel regulation in hippocampal neurons. In this study, the authors show that carvedilol and alprenolol can promote Gs activity and PKA phosphorylation of β-adrenergic receptor and LTCC at Ser1928, and increased LTCC activity. The authors contend that some β blockers can activate beta2 adrenergic receptors and target specific downstream LTCCs, and could explain the effect of β blockers on the central nervous system.

Essential revisions:

1) In part, the manuscript incrementally builds upon several other studies, including these authors, probing the concept that some β adrenergic antagonists have agonist properties, and that in hippocampus, the LTCC 's may be positioned close to beta2 adrenergic receptors and are a "privileged" downstream signaling target. Please clearly indicate why this study should not be considered incremental.

2) The conclusion at the end of the manuscript attempting to create relevance to the findings, in regards to neuronal toxicities, is overstated without data showing that this local signaling could have these effects. Having data showing neuronal toxicities that are both carvedilol and Ca^2+^ dependent would have made the manuscript more than more substantial. Are there data showing neuronal toxicities (not just known reversible side effects associated with β blockers) associated with carvedilol in humans? Otherwise, this angle should be greatly downplayed.

3) All reviewers felt that the other major conclusion that this β-blocker signaling is highly spatially restricted to only the local domain of the B2AR receptor is not quite as well supported and could benefit from some additional experiments, which I would prioritize as: 1) evaluating whether CAR is only effective in regulation L-channel currents when applied through the patch pipet as shown in Figure 4 but *not* when applied in the bath and 2) using a generic plasma membrane targeted ICUE3 cAMPs sensor to better characterize how local the CAR and ALP generated cAMP signals are compared to the full-agonist ISO. The authors already show that CAR regulation of Ca_v_1.2 S1928-P and S1700-P is block by the B2AR-specific antagonist ICI.

4) The authors use BayK to enhance channel activities and increase open time. This is a suboptimal method to establish that phosphorylation has a significant effect on signaling in native conditions. To convincingly show that LTCC are indeed a bona fide target of carvedilol, the experiments should be performed without BayK so that we can determine the actual effect of beta2-adrenergic receptor activation by carvedilol. The effects of PKA phosphorylation of LTCC may or may not be the same in the absence of BayK. The effects of carvedilol should be tested, at least on as subset of experiments, without BayK. The impact of BayK on channel modulation should be addressed. It is not necessary to use BayK to measure calcium channel activity.

5) Concerns about the variability of NPo. For instance, In Figure 6C, the data are driven by 7 points. In fact, the majority of data-points are similar to mutant and mutant + Iso. This variability suggest that the authors should develop a system so that they can measure the basal current and then infuse the drug via the patch pipette. Without understanding the number of channels in each patch and the basal open probability, the experiments have limited value. The single channel recordings clearly indicate bimodal gating, i.e. periods of low and high open probability. How do others authors take this into account? One would need very many long recordings to do this. This issue is further confounded by using BayK to artificially elevate channel open probability.

Statistics:

In some Figures such as 3A, 4E, 6C, the n's for carvedilol are much more than other groups. Why were the experiments designed in such a way with a disproportional bias towards a particular compound? The details of replicates were not given in the figure legends. For example, how many separate experiments have been performed? And in the separate experiments, were all the different groups represented? What is the rationale for the disproportional representation among different groups?

---

## [Author Response]

Essential revisions:1) In part, the manuscript incrementally builds upon several other studies, including these authors, probing the concept that some β adrenergic antagonists have agonist properties, and that in hippocampus, the LTCC 's may be positioned close to beta2 adrenergic receptors and are a "privileged" downstream signaling target. Please clearly indicate why this study should not be considered incremental.

We thank the reviewers for the comments. We now have revised the Abstract and Discussion section of our manuscript to include an expanded description of the significance of our study. (1) The concept that some β-blockers have agonist properties was only observed at high drug concentrations and in vitro with reconstituted systems before, our study for the first time show that β-blockers can activate receptor at nanomolar concentrations in living cells, which is clinically relevant in contrast to superphysiological concentrations in previous studies. In detail, as low as 1 nM alprenolol as well as 1 nM carvedilol induced 20-40% of maximal effects (as obtained with 1 μM isoproterenol) with respect to phosphorylation of β_2_AR by PKA and to cAMP production detected by the ICUE3 sensor coupled to β_2_AR. (2) Unlike agonists triggering both PKA and GRK pathways, activation of β_2_AR by β-blockers selectively transduce cAMP/PKA signaling, but not GRK signaling, function as PKA-biased drugs. This is a phenomenon never reported before, especially for carvedilol which was recognized as a GRK-biased drug in the past. (3) Importantly, we show that β-blocker-induced cAMP/PKA signaling can differentiate two primary downstream targets (LTCC vs. AMPAR) in primary neurons, which further exemplifies biased actions as well as unique tools to dissect these two functionally relevant ion channels in neuronal physiology in future. The differential signaling by carvedilol with respect to LTCC and AMPAR is especially remarkable because both channels form complexes with β_2_AR that are localized within dendritic spines (the postsynaptic sites of glutamatergic synapses), which are only ~500 nm in diameter.

2) The conclusion at the end of the manuscript attempting to create relevance to the findings, in regards to neuronal toxicities, is overstated without data showing that this local signaling could have these effects. Having data showing neuronal toxicities that are both carvedilol and Ca^2+^ dependent would have made the manuscript more than more substantial. Are there data showing neuronal toxicities (not just known reversible side effects associated with β blockers) associated with carvedilol in humans? Otherwise, this angle should be greatly downplayed.

We thank the reviewers for these insightful comments. Now, we have added cell viability assay (Figure 5—figure supplement 2) showing neuronal toxicity induced by carvedilol, which can be reversed by the blockage of β_2_AR or LTCC.

3) All reviewers felt that the other major conclusion that this β-blocker signaling is highly spatially restricted to only the local domain of the B2AR receptor is not quite as well supported and could benefit from some additional experiments, which I would prioritize as: 1) evaluating whether CAR is only effective in regulation L-channel currents when applied through the patch pipet as shown in Figure 4 but not when applied in the bath and 2) using a generic plasma membrane targeted ICUE3 cAMPs sensor to better characterize how local the CAR and ALP generated cAMP signals are compared to the full-agonist ISO. The authors already show that CAR regulation of Ca_v_1.2 S1928-P and S1700-P is block by the B2AR-specific antagonist ICI.

We thank the reviewers for these suggestions and now have added the two suggested experiments, which support the original conclusions.

(1) Now in Figure 5A-5E and Figure 5—figure supplement 1C-1D, we show that carvedilol is only effective in regulation L-type channel currents when applied through the patch pipette but not when perfused in the bath, regardless of whether BayK was present or not in the pipette solution.

(2) Now in Figure 3—figure supplement 3, we have tested two generic PM targeted ICUE3 sensors and show that the carvedilol and alprenolol generated cAMP signals are highly restricted in the vicinity of the activated receptors when compared to the full-agonist ISO.

4) The authors use BayK to enhance channel activities and increase open time. This is a suboptimal method to establish that phosphorylation has a significant effect on signaling in native conditions. To convincingly show that LTCC are indeed a bona fide target of carvedilol, the experiments should be performed without BayK so that we can determine the actual effect of beta2-adrenergic receptor activation by carvedilol. The effects of PKA phosphorylation of LTCC may or may not be the same in the absence of BayK. The effects of carvedilol should be tested, at least on as subset of experiments, without BayK. The impact of BayK on channel modulation should be addressed. It is not necessary to use BayK to measure calcium channel activity.

Although we agree that it is not always necessary to use BayK, this L-type channel agonist has been traditionally used in this type of single-channel recordings because it promotes channel activity and longer open-times. This facilitates the detection and measurement of unitary channel openings and increases the proportion of sweeps with activity, a maneuver that has been extensively employed in prior studies to facilitate the examination of single L-type channel activity [1-8]. Note that BayK is present in all experimental conditions, including NT. Therefore, any change in channel activity is expected to result from the effects of the stimuli provided, be that ISO or carvedilol. Consistent with this, our data here showing that ISO increases the open probability of L-type channels are consistent with a wealth of studies in the literature [6,9-14], thus suggesting that the presence of BayK did not mask the ISO effects.

Nonetheless, as requested, we repeated a subset of the single-channel experiments without BayK in the patch pipette (Figure 5—figure supplement 1). Under these conditions and in our hands, the probability of observing patches with single-channel activity in neurons was greatly reduced. Indeed, patches with >5% channel availability within the first 50 sweeps that could be used for analysis accounted for ≤30% of all patches recorded (Figure 5—figure supplement 1A). Note that these are challenging experiments and that the reduced frequency of detecting events without BayK in the pipette solution delays an already very time consuming experimental process. Figure 5—figure supplement 1 shows all the data that we were able to collect within the two-month period given for resubmission. From patches with single-channel activity, we found that data acquired without BayK in the patch pipette resembled those with BayK (see Figure 5—figure supplement 1 and Figure 5). Importantly, in carvedilol backfilling pipette experiments, L-type channel activity was increased regardless of whether or not BayK was present in the patch pipette (see Figure 5F-G and Figure 5—figure supplement 1C). These new data support our hypothesis that L-type channels are a downstream target of β_2_AR activation by carvedilol.

5) Concerns about the variability of NPo. For instance, In Figure 6C, the data are driven by 7 points. In fact, the majority of data-points are similar to mutant and mutant + Iso. This variability suggest that the authors should develop a system so that they can measure the basal current and then infuse the drug via the patch pipette. Without understanding the number of channels in each patch and the basal open probability, the experiments have limited value. The single channel recordings clearly indicate bimodal gating, i.e. periods of low and high open probability. How do others authors take this into account? One would need very many long recordings to do this. This issue is further confounded by using BayK to artificially elevate channel open probability.

We agree that the variability of nPo can be a critical element to interpret single channel data due to overstating open probability based on a high n number. Therefore, we corrected our parameter by the number of channels (*n*) and determined in addition to open probability slow and fast kinetic and the mean ensemble average current. We further performed backfilling experiments in which the pipette tip was filled by capillary force with control solution and carvedilol was loaded with a hypodermic needle from the back end of the pipette. This configuration allows measuring basal L-type channel activity immediately after seal formation before the carvedilol effects could take placed. These carvedilol backfilling experiments show that LTCC activity was relatively low at the beginning of the recording but significantly increased as the drug diffused to the pipette tip. Similar results were observed regardless of whether BayK was included or not in the pipette solution (Figure 5F-G and Figure 5—figure supplement 1C). These new data provide further support to our original observation that carvedilol increases LTCC activity in neurons.

We agree with the reviewers that carvedilol seems to have a bimodal effect with some patches showing increased L-type channel activity and others displaying no or minimal effect compared to NT recordings. Whether this is a reflection of or related to bimodal L-type channel gating is unclear [15]. Our new FRET analysis show that carvedilol-induced cAMP signals are highly restricted within the receptor local domain and cannot be detected by the cAMP biosensors targeted to lipid rafts or non-lipid rafts in the plasma membrane; this finding is in contrast to the broad distribution of cAMP signals induced by ISO. These data indicate that only the channels in the complex with activated β_2_AR will display increased activity after carvedilol stimulation. In some patches, LTCCs without β_2_AR in the complexes won’t display increases in channel activities after stimulation with carvedilol. Nonetheless, these results are reminiscent and in agreement with earlier studies describing β adrenergic effects on L-type channels [10-11,16-19]. We plan to examine this possibility in subsequent research because, as the reviewers well pointed out, addressing this issue will require a substantial number of additional experiments that fall beyond the scope of this study, are unlikely to change the conclusions, and these experiments will require a much longer time frame.

Statistics:In some Figures such as 3A, 4E, 6C, the n's for carvedilol are much more than other groups. Why were the experiments designed in such a way with a disproportional bias towards a particular compound? The details of replicates were not given in the figure legends. For example, how many separate experiments have been performed? And in the separate experiments, were all the different groups represented? What is the rationale for the disproportional representation among different groups?

Now the repeats of separate experiments and n’s for each condition are clearly noted in figure legends for all figures.

For Figure 3A (which is Figure 3A now), we apologize for not stating clearly in figure legend – we repeated separate experiments three times. Now we have combined data of three repeats together, and the n's for carvedilol are similar to other groups.

For the single-channel data in Figures 4E and 6C (which are Figures 5C and 7E now), we repeated the experiments for all conditions using at least three different neuronal cultures. For all the conditions, except carvedilol, we found that the single-channel data had a tight spread, even if they did not follow a normal distribution, which was consistent with prior results from our group [13-14,20]. When attempting the carvedilol experiments for the first time, however, we immediately noticed a broad distribution in the data with some patches displaying low channel activity and others high channel activity. This prompted us to collect as much data as possible for this particular experimental condition. Because we did not find a rational scientific justification to exclude a particular result, we found it necessary to include all the data collected in our analysis. As we discussed above, this broad (or bimodal) distribution in LTCC activity in response to carvedilol could be due to whether β_2_AR-LTCC complexes are present in the patches. We will only observe carvedilol-induced increases in channel activities when β_2_AR is present in the patch; and some patches may lack the complex despite the presence of LTCCs.

References:

1) Hess P, Lansman JB, Tsien RW. Calcium channel selectivity for divalent and monovalent cations. Voltage and concentration dependence of single channel current in ventricular heart cells. J Gen Physiol. 1986; 88:293-319

2) Schuhmann K, Romanin C, Baumgartner W, Groschner K. Intracellular Ca^2+^ inhibits smooth muscle L-type Ca^2+^ channels by activation of protein phosphatase type 2b and by direct interaction with the channel. J Gen Physiol. 1997; 110:503-513

3) Costantin J, Noceti F, Qin N, Wei X, Birnbaumer L, Stefani E. Facilitation by the beta2a subunit of pore openings in cardiac Ca^2+^ channels. J Physiol. 1998; 507 (1):93-103

4) Yue DT, Marban E. Permeation in the dihydropyridine-sensitive calcium channel. Multi-ion occupancy but no anomalous mole-fraction effect between ba^2+^ and Ca^2+^. J Gen Physiol. 1990; 95:911-939.

5) Wang SQ, Song LS, Lakatta EG, Cheng H. Ca^2+^ signalling between single L-type Ca^2+^ channels and ryanodine receptors in heart cells. Nature. 2001; 410:592-596

6) Davare MA, Avdonin V, Hall DD, Peden EM, Burette A, Weinberg RJ, Horne MC, Hoshi T, Hell JW. A beta2 adrenergic receptor signaling complex assembled with the Ca^2+^ channel Ca_v_1.2. Science. 2001; 293:98-101

7) Dzhura I, Neely A. Differential modulation of cardiac Ca^2+^ channel gating by beta-subunits. Biophys J. 2003; 85:274-289

8) Navedo MF, Amberg GC, Votaw VS, Santana LF. Constitutively active L-type Ca^2+^ channels. Proc Natl Acad Sci U S A. 2005; 102:11112-11117

9) Artalejo CR, Ariano MA, Perlman RL, Fox AP. Activation of facilitation calcium channels in chromaffin cells by d1 dopamine receptors through a camp/protein kinase a-dependent mechanism. Nature. 1990; 348:239-242

10) Yue DT, Herzig S, Marban E. Beta-adrenergic stimulation of calcium channels occurs by potentiation of high-activity gating modes. Proc Natl Acad Sci U S A. 1990; 87:753-757.

11) Schroder F, Herzig S. Effects of beta2-adrenergic stimulation on single-channel gating of rat cardiac L-type Ca^2+^ channels. Am J Physiol. 1999; 276:834-843

12) Erxleben C, Liao Y, Gentile S, Chin D, Gomez-Alegria C, Mori Y, Birnbaumer L, Armstrong DL. Cyclosporin and timothy syndrome increase mode 2 gating of Ca_v_1.2 calcium channels through aberrant phosphorylation of S6 helices. Proc Natl Acad Sci U S A. 2006; 103:3932-3937

13) Patriarchi T, Qian H, Di Biase V, Malik ZA, Chowdhury D, Price JL, Hammes EA, Buonarati OR, Westenbroek RE, Catterall WA, Hofmann F, Xiang YK, Murphy GG, Chen CY, Navedo MF, Hell JW. Phosphorylation of Ca_v_1.2 on S1928 uncouples the L-type Ca^2+^ channel from the beta2 adrenergic receptor. EMBO J. 2016; 35:1330-1345

14) Shen A, Nieves-Cintron M, Deng Y, Shi Q, Chowdhury D, Qi J, Hell JW, Navedo MF, Xiang YK. Functionally distinct and selectively phosphorylated gpcr subpopulations co-exist in a single cell. Nat Commun. 2018; 9:1050

15) Hess P, Lansman JB, Tsien RW. Different modes of Ca channel gating behaviour favoured by dihydropyridine Ca agonists and antagonists. Nature. 1984; 311:538-544

16) Herzig S, Patil P, Neumann J, Staschen CM, Yue DT. Mechanisms of beta-adrenergic stimulation of cardiac Ca^2+^ channels revealed by discrete-time markov analysis of slow gating. Biophys J. 1993;65:1599-1612

17) Xiao RP, Lakatta EG. Beta 1-adrenoceptor stimulation and beta 2-adrenoceptor stimulation differ in their effects on contraction, cytosolic Ca^2+^, and Ca^2+^ current in single rat ventricular cells. Circ Res. 1993; 73(2):286-300.

18) Zhou YY, Cheng H, Bogdanov KY, Hohl C, Altschuld R, Lakatta EG, Xiao RP. Localized cAMP-dependent signaling mediates beta 2-adrenergic modulation of cardiac excitation-contraction coupling. Am J Physiol. 1997; 273:1611-1618.

19) Klein G, Drexler H, Schroder F. Protein kinase G reverses all isoproterenol induced changes of cardiac single L-type calcium channel gating. Cardiovasc Res. 2000; 48:367-374

20) Qian H, Patriarchi T, Price JL, Matt L, Lee B, Nieves-Cintron M, Buonarati OR, Chowdhury D, Nanou E, Nystoriak MA, Catterall WA, Poomvanicha M, Hofmann F, Navedo MF, Hell JW. Phosphorylation of Ser1928 mediates the enhanced activity of the L-type Ca^2+^ channel Ca_v_1.2 by the beta2-adrenergic receptor in neurons. Science signaling. 2017;10:eaaf9659